# Short-stay urgent hospital admissions of children with convulsions: A mixed methods exploratory study to inform out of hospital care pathways

Cari Malcolm[1], Pat Hoddinott[2]*, Emma King[2], Smita Dick[3], Richard Kyle[4], Philip Wilson[5,6], Emma France[2], Lorna Aucott[7], Stephen W. Turner[3,8]

1 School of Health Sciences, University of Dundee, Dundee, United Kingdom, 2 Nursing, Midwifery and Allied Health Professions Research Unit, University of Stirling, Stirling, United Kingdom, 3 Child Health, University of Aberdeen, Aberdeen, United Kingdom, 4 Academy of Nursing, Department of Health and Care Professions, Faculty of Health and Life Sciences, University of Exeter, Exeter, United Kingdom, 5 Centre for Rural Health, University of Aberdeen, Aberdeen, United Kingdom, 6 Centre for Research and Education in General Practice, University of Copenhagen, København, Denmark, 7 Health Services Research Unit, University of Aberdeen, Aberdeen, United Kingdom, 8 Women and Children Division, NHS Grampian, Aberdeen, United Kingdom

* p.m.hoddinott@stir.ac.uk

## Abstract

### Objective

To inform interventions focused on safely reducing urgent paediatric short stay admissions (SSAs) for convulsions.

### Methods

Routinely acquired administrative data from hospital admissions in Scotland between 2015–2017 investigated characteristics of unscheduled SSAs (an urgent admission where admission and discharge occur on the same day) for a diagnosis of febrile and/or afebrile convulsions. Semi-structured interviews to explore perspectives of health professionals (n = 19) making referral or admission decisions about convulsions were undertaken. Interpretation of mixed methods findings was complemented by interviews with four parents with experience of unscheduled SSAs of children with convulsion.

### Results

Most SSAs for convulsions present initially at hospital emergency departments (ED). In a subset of 10,588 (11%) of all cause SSAs with linked general practice data available, 72 (37%) children with a convulsion contacted both the GP and ED pre-admission. Within 30 days of discharge, 10% (n = 141) of children admitted with afebrile convulsions had been readmitted to hospital with a further convulsion. Interview data suggest that panic and anxiety, through fear that the situation is life threatening, was a primary factor driving hospital attendance and admission. Lengthy waits to speak to appropriate professionals exacerbate

**Data Availability Statement:** The data collected and analysed during this study are not publicly available due to privacy issues. We are not able to

share this data as they contain potentially identifying and sensitive participant information. Moreover, participants were not informed and did not give consent for their transcripts to be shared in a public repository. Queries regarding data access may be directed to the authors or the research governance team (researchgovernance@abdn.ac.uk).

**Funding:** This study was funded by the Chief Scientist Office, Scottish Government Health and Social Care Directorate (Award Number HIPS/18/09). The funders had no role in study design, data collection and analysis, decision to publish, or preparation of the manuscript.

**Competing interests:** The authors have declared that no competing interests exist.

parental anxiety and can trigger direct attendance at ED, whereas some children with complex needs had direct access to convulsion professionals.

## Conclusions

SSAs for convulsions are different to SSAs for other conditions and our findings could inform new efficient convulsion-specific pre and post hospital pathways designed to improve family experiences and reduce admissions and readmissions.

## Introduction

Convulsions, febrile and afebrile, are a common reason for children's emergency department (ED) attendance and short-stay admissions (SSA) to hospital [1–4], placing considerable burden on health care systems as well as children and families. Febrile and afebrile convulsions are often similar in presentation, both being generally brief and resolving spontaneously prior to assessment by a health professional [5]. The first occasion a child has loss of consciousness and involuntary movements associated with a convulsion will usually prompt an ambulance call and be conveyed to the nearest ED. Assuming that the convulsions are not complex in nature, that the child has recovered completely, and that parental reassurance, education, and safety netting has been provided, hospitalisation is seldom required and any further convulsions can usually be safely managed at home or in the community [5–8]. High levels of ED attendance and SSAs could indicate poor management of the condition, challenges in accessing specialist care, or that parents feel unable to manage a convulsion at home [9–10].

The **Fl**ow of **Adm**issions in Ch**i**ldren and You**ng** Pe**o**ple (FLAMINGO) project was undertaken in response to the growing number of paediatric urgent hospital admissions in the UK [4–11], explained largely by rising SSAs [4, 12]. This exploratory sequential mixed-methods study aimed to improve understanding of SSAs in Scotland through linkage of national databases to identify referral pathways and characteristics of SSAs. Qualitative interviews with professionals were complemented by parent experiences and Public Patient Involvement. The main FLAMINGO study findings, published elsewhere, report on quantitative data linkage analysis of routinely acquired Scottish data to identify referral sources and characteristics for urgent paediatric SSAs arising from a range of common conditions including, but not limited to, viral illness, asthma and bronchiolitis [13, 14] and on qualitative interviews with 21 parents and 48 health professionals communicating their desired outcomes of pre-hospital urgent care for these urgent SSAs [15]. Data linkage findings from the main FLAMINGO dataset indicated a high prevalence of SSAs for convulsions presenting at the emergency department (ED) [13] which, given the lack of recently published evidence in this area, warranted further detailed investigation. Previous FLAMINGO papers [13–16] did not specifically report the characteristics of SSAs related to convulsions or parental and professional experiences of these admissions. Therefore, additional quantitative analysis and qualitative research involving the recruitment of Epilepsy Specialist Nurses (ESN)s was undertaken. This paper communicates novel findings from this research to identify learning for improvement in care pathways and development of potential interventions focused on safely reducing urgent SSAs for convulsions and improving care pathways, by analysing (i) characteristics of SSAs for children with afebrile or febrile convulsions, and (ii) qualitative interviews with professionals and parents with experience of childhood convulsions.

## Methods

### Quantitative data linkage

A comprehensive account of the data linkage methodology has been published [13] and is summarised here. Anonymised details of all urgent medical paediatric (0 to <16 years of age) hospital admissions and SSAs in Scotland between 2015 and 2017 were identified from the Scottish Morbidity Record 01 database (SMR01). These were linked with the Unscheduled Care Datamart (UCD) and general practice data obtained by the Trusted Third Party agent Albasoft. GP data were available after the data custodian opted in to share data, and this occurred for 11% of admissions.

All unscheduled and SSAs for a diagnosis of febrile and/or afebrile convulsions, according to the International Classification of Diseases, 10th Revision (ICD-10) categories of R560, G409, and R568 were examined to identify trends in the following factors of interest: child characteristics (age, sex, ethnicity, SIMD); admission characteristics (time of day, day of week, month of year, referral source). Definitions related to this paper, and specifically the data linkage methodology, are provided in Table 1.

Descriptive quantitative data are presented. Characteristics of SSAs for convulsion and all other SSAs were compared using Chi squared test, Student's t-test and Mann-Whitney U test as appropriate. Standard statistical software was used (IBM SPSS version 24.0.0.0).

### Qualitative interviews

Our sampling approach and methods for conducting qualitative interviews with parents and professionals are reported in Malcolm et al. [15]. To focus more specifically on the experience of SSAs for childhood convulsions, and complement the existing qualitative FLAMINGO dataset, additional recruitment of paediatric Epilepsy Specialist Nurses (ESNs) was undertaken. ESNs working in five Health Boards, selected for the diversity observed in the quantitative findings [14], were purposively sampled. Potential ESN participants were sent an information sheet, study invitation, and contact details for the research team to opt into a telephone interview.

Telephone interviews with ESNs were conducted between January and March 2021 by researchers (CM, EK) with experience of qualitative health services research (CM, EK) and clinical paediatric nursing (CM). The original FLAMINGO study semi-structured topic guide was modified, once data linkage findings for convulsions were known, to include specific questions about experiences of SSAs for convulsions (S1 File). All interviews were audio recorded, anonymised, transcribed verbatim, and uploaded to QSR International NVivo (version 12) software for data management and analysis of data relating to convulsions.

The FLAMINGO qualitative dataset, comprising parent interviews for all causes of SSAs, included four parents discussing their experiences of SSAs for convulsions. Their experiences contribute to the interpretation of data linkage and professional interview findings.

Thematic framework analysis [17, 18] was completed to explore the experiences of unscheduled SSAs of children with convulsion. Members of the FLAMINGO qualitative team (EK, CM, EF, PH) familiarised themselves with the data, independently read a sample of four transcripts, two from each participant group (parents and professionals) and met to discuss and develop an initial high-level coding framework. The analytical coding framework was systematically applied to the remaining transcripts. EK and CM carried out line by line coding, with weekly discussion with PH and EF. The team agreed final themes, data mapping and interpretation using a framework matrix to organise data characteristics for parents and professionals facilitating cross-Health Board and within-sample comparisons. To ensure rigour and trustworthiness

**Table 1. Definitions of key terms referred to in this paper.**

| | |
|---|---|
| **Albasoft (GP data)** | General Practice (GP) data were made available through an NHS Trusted Third Party (Albasoft). |
| **ICD-10 term and code** | Febrile convulsions R560: febrile convulsions; Afebrile convulsions G409: epilepsy, unspecified; R568: other and unspecified convulsions |
| **Health Board** | Health Boards are geographical areas in Scotland where healthcare is administered to the population by a single organisation. There are 11 mainland Health Boards in Scotland and three Island Health Boards with no inpatient facilities for children. Two of the 11 mainland Health Boards have two separate inpatient facilities for children (NHS Grampian and NHS Lothian). Two hospitals with inpatient facilities have dedicated children's emergency departments staffed by paediatricians (NHS Greater Glasgow and Clyde and NHS Lothian). |
| **Primary Care Out of Hours (OOH) service** | The Primary Care Out of Hours (OOH) service provides urgent care in the community for those who need to be seen by a General Practitioner (GP) when their GP practice is closed. In Scotland this is typically between 18:00 and 08:00am on weekdays, all weekends and on public holidays. |
| **NHS24** | NHS24 forms part of the unscheduled care services in Scotland, providing a telephone triage service (accessed through dialling 111) and urgent health advice out of office hours when there is no access to GP and primary care services. Non-clinical call handlers will ask protocol-guided questions and then transfer the individual to the most relevant healthcare professional, such as a nurse practitioner or pharmacy advisor. The healthcare professional will typically take one of the following actions: provide information and general healthcare advice; refer and schedule an appointment with an out of hours GP (OOH); or advise ED attendance. |
| **Readmission** | Where two urgent admissions occur within 30 days. In this paper we report readmissions for any reason, and also readmissions with the same diagnosis. |
| **Referral source** | The location (ED, GP, OOH or combination thereof) where the child had a clinical record on the same day of admission. |
| **Scottish Index of Multiple Deprivation (SIMD)** | The SIMD is a standardised tool for classifying geographical areas of Scotland according to their relative level of deprivation. An index of deprivation is created across seven domains: income, employment, education, health, access to services, crime, and housing. Deprivation is categorised into quintiles from SIMD1 (most deprived) to SIMD5 (least deprived). The 2016 (5th) version is used in this project. |
| **Short stay admission (SSA)** | SSA is an urgent admission where a child is admitted and discharged on the same day. |
| **Scottish Morbidity Record 01 (SMR01)** | The SMR01 database, made available by the NHS National Services Scotland Information Services Division (ISD), records episodes of day-case or inpatient care in the Scottish NHS hospitals (excluding neonatal, maternity, or psychiatric settings). Data provided include sex; ethnic group; decimal age; date of admission; date of discharge; admission type (emergency or elective); SIMD quintile; up to six ICD-10 diagnoses; Health Board of admission; and specialty (paediatric medicine, surgery, or dentistry). |
| **Unscheduled Care Datamart (UCD)** | This resource provides data from ED, OOH, and NHS24 (including the Scottish Ambulance Service). |
| **Urgent admission** | An unplanned (or emergency) admission to a medical paediatric inpatient facility. This does not include cases seen in and then discharged directly from the Emergency Department (ED). |
| **Out of hours** | Out of hours services refer to any access to care outside typical 'working hours', when general practices are closed. In Scotland this is typically between 18:00 and 08:00am on weekdays, all weekends and on public holidays. |

we remained iterative, open, and reflexive throughout the analysis keeping notes and an audit trail from the raw data to the finalised themes [18]. Integration of data-linkage and the qualitative findings were discussed at monthly meetings of the entire FLAMINGO team.

### Ethics approval and ethical considerations

The quantitative part of the study was approved by the National Caldicott guardian (PBPP 1718–0836) and the quantitative part by the NHS North of Scotland Research Ethics Service (REC references: 19/NS/0134). All participants provided written consent to take part in the study.

## Results

### Unscheduled urgent paediatric hospital admissions for febrile and afebrile convulsions

Between 2015 and 2017, there were 171,039 urgent medical paediatric admissions to Scottish hospitals, 92,229 (54%) of which were SSAs [13]. A referral source was identified for 45,873 (49.7%) of these SSAs, however GP data were only available for 10,588 (11%). Febrile convulsions accounted for 1,916 (1.1%) of admissions, including 976 (1.6%) SSAs, whilst afebrile convulsions accounted for 2,661 (1.6%) of admissions, including 1,344 (1.5%) SSAs. Relevant pre-existing conditions were identified for 284 (21%) of individuals admitted with afebrile convulsions and 22 (2%) with febrile convulsions (list of pre-existing conditions included in S2 File).

The ED was the referral source for 625 (64%) of SSAs for febrile convulsions and for 830 (62%) of afebrile convulsions (Table 2). The referral source for 211 admissions with convulsions within the subset where GP data were available was: GP 22 (10%); OOH 7 (3%); ED 105 (50%); and more than one referral source 77 (37%), of whom 72 had seen both GP and ED on the same day. In this subset, the proportions of referral sources for conditions other than convulsions were GP 3362 (42%); OOH 816 (10%); ED 1748 (22%); and 2088 (26%) for more than one source.

We have previously reported that children where ED was the referral source for all conditions leading to SSAs were older (median age 2.4), more likely to be from the most deprived communities (n = 9,440; 32%) and to have a white European ethnicity (n = 21,360; 73%) [13]. Table 3 compares this all SSA data with data for febrile, afebrile and all convulsions. The median (interquartile range) age of all children admitted with convulsions was 2.8 years (1.5, 6.8) compared to 2.2 years (0.7, 5.7) for other SSAs, p<0.001. The majority of children admitted with febrile convulsions were under four years of age. Children admitted with afebrile convulsions were spread evenly across the age groups. The proportion of White European children admitted with febrile and afebrile convulsions and comparable to SSA admissions referred from ED. The proportions of SSAs in each SIMD quintile for both febrile and afebrile

**Table 2. Final referral source for paediatric SSAs of febrile (n = 976) and afebrile convulsions (n = 1344).**

| Referral Source | Febrile Convulsion n (%) | Afebrile Convulsion n (%) | All convulsions n (%) | All SSAs n (%) |
|---|---|---|---|---|
| GP† | 19 (2.0) | 22 (1.6) | 41 (1.7) | 3384 (3.7) |
| OOH | 22 (2.3) | 20 (1.5) | 42 (1.8) | 7569 (8.2) |
| ED | 625 (64.0) | 830 (61.8) | 1455 (62.7) | 29,461 (31.9) |
| More than one referrer* | 93 (9.5) | 79 (5.9) | 172 (7.4) | 5459 (5.9) |
| Unknown/Missing Data† | 217 (22.2) | 393 (29.2) | 610 (26.3) | 46356 (50.2) |
| All referral sources | 976 | 1344 | 2320 | 92,229 |

†GP data were only available for 11% (n = 10,588) of all SSAs (n = 92,229).

*More than one referrer with a combination of services making up the referral source including GP and OOH; GP and ED; OOH then ED; ED then OOH; GP then OOH then ED; GP then ED then OOH.

**Table 3. Characteristics of paediatric SSAs for febrile and afebrile convulsions.**

| Characteristic: n (%) | Febrile Convulsion (n = 1344) | Afebrile Convulsion (n = 1344) | All convulsions (n = 2320) | All SSAs* (n = 92229) | All SSAs with ED referral source (n = 29461) | P value comparing all convulsions with all SSAs |
|---|---|---|---|---|---|---|
| **Sex:** | | | | | | |
| Female | 424 (43.4) | 633 (47.1) | 1057 (46) | 40728 (44.2) | 12976 (44.0) | 0.168 |
| Male | 552 (56.6) | 711 (52.9) | 1263 (54) | 51501 (55.8) | 16485 (56.0) | |
| **Age category at admission (years):** | | | | | | |
| < 1 | 96 (9.8) | 131 (9.8) | 227 (10.0) | 28087 (30.5) | 7592 (25.8) | <0.001 |
| 1–4 | 818 (83.8) | 499 (37.1) | 1317 (56.6) | 37644 (40.8) | 13282 (45.1) | |
| 5–9 | 54 (5.5) | 387 (28.8) | 441 (19.0) | 14480 (15.7) | 4622 (15.7) | |
| 10–15 | 5 (0.5) | 323 (24.0) | 328 (14.1) | 11729 (12.8) | 3965 (13.5) | |
| Unknown/Missing Data | 3 (0.3) | 4 (0.3) | 7 (0.3) | 289 (0.3) | | |
| **Ethnicity:** | | | | | | |
| White European | 684 (70.0) | 1057 (78.6) | 1741 (75.0) | 63029 (68.4) | 21360 (72.5) | <0.001 |
| Other | 292 (30.0) | 287 (21.4) | 579 (25.0) | 29200 (31.6) | 8101 (27.5) | |
| **SIMD:** | | | | | | |
| SIMD 1 (most deprived) | 253 (26.0) | 383 (28.5) | 636 (27.8) | 25022 (27.4) | 9440 (32.0) | 0.014 |
| SIMD 2 | 215 (22.0) | 321 (24.0) | 536 (23.4) | 20118 (22.1) | 6685 (22.7) | |
| SIMD 3 | 215 (22.0) | 257 (19.1) | 472 (20.6) | 17612 (19.3) | 5051 (17.1) | |
| SIMD 4 | 160 (16.4) | 195 (14.5) | 355 (15.5) | 16405 (18.0) | 4540 (15.4) | |
| SIMD 5 (least deprived) | 121 (12.4) | 170 (12.6) | 291 (12.7) | 12032 (13.2) | 3539 (12.0) | |
| Unknown/Missing Data | 12 (1.2) | 18 (1.3) | 30 (1.2) | 1040 (1.1) | 206 (0.7) | |
| **Day of Week:** | | | | | | |
| Monday | 141 (14.4) | 207 (15.4) | 348 (15.0) | 15196 (16.5) | 4769 (16.2) | <0.001 |
| Tuesday | 111 (11.4) | 190 (14.1) | 301 (13.0) | 13397 (14.5) | 4485 (15.2) | |
| Wednesday | 144 (14.8)) | 225 (16.7) | 369 (15.9) | 14234 (15.4) | 4528 (15.4) | |
| Thursday | 149 (15.3) | 211 (15.7) | 360 (15.5) | 14589 (15.8) | 4698 (15.9) | |
| Friday | 126 (12.9) | 188 (14.0) | 314 (13.5) | 14256 (15.5) | 4536 (15.4) | |
| Saturday | 176 (18.0) | 161 (12.0) | 337 (14.5) | 10004 (10.8) | 3142 (10.7) | |
| Sunday | 129 (13.2) | 162 (12.0) | 291 (12.5) | 10553 (11.4) | 3303 (11.2) | |
| **Time of arrival in ED:** | | | | | | |
| 00:00–06:00 | 211 (21.6) | 189 (14.1) | 400 (24.8) | 6577 (7.1) | 5784 (19.6) | <0.001 |
| 06:01–12:00 | 137 (14.0) | 336 (25.0) | 473 (29.4) | 9656 (10.5) | 8380 (28.4) | |

*(Continued)*

**Table 3.** (Continued)

| Characteristic: n (%) | Febrile Convulsion (n = 1344) | Afebrile Convulsion (n = 1344) | All convulsions (n = 2320) | All SSAs* (n = 92229) | All SSAs with ED referral source (n = 29461) | P value comparing all convulsions with all SSAs |
|---|---|---|---|---|---|---|
| 12:01–18:00 | 271 (27.8) | 309 (23.0) | 580 (36.0) | 14,508 (15.7) | 12377 (42.0) | |
| 18:01–24:00 | 88 (9.0) | 70 (5.2) | 158 (9.8) | 3948 (4.3) | 292 (9.9) | |
| Not admitted via ED | 296 (27.6) | 440 (32.7) | 736 (31.7) | 57,540 (62.4) | 62,768 (68.1) | |
| **Arrival in ED or OOH during out of hours** (Includes Monday-Friday between 18:01–08:00 or at the weekend) | 262 (26.8) | 244 (18.2) | 506 (21.8) | 16375 (17.8) | 6447 (22.0) | <0.001 |
| **Readmissions within 30 days** | 72 (7.43) | 249 (18.5) | 321 (13.8) | 12378 (13.4) | 3145 (10.7) | 0.552 |
| **Readmissions within 30 days for the same reason** | 42 (4.3) | 141 (10.5) | 183 (7.9) | 5976 (6.5) | 1265 (4.3) | 0.005 |

*Full reporting of all SSAs for the 10 most common composite diagnoses is reported in Dick et al. [13]

convulsions were comparable; 636 (27.8%) of all convulsion related SSAs lived in the most deprived quintile compared to 9940 (32.0%) for all ED admissions.

Admissions with convulsions were more evenly distributed across days of the week compared to both all SSAs and all SSAs with ED as a referral source. Relatively few convulsions attended ED between 6pm and midnight, with proportions similar to all SSA referred via ED.

Readmissions within 30 days after discharge following a SSA were more common among children with afebrile convulsions (n = 249; 18.5%) compared to both febrile convulsions (n = 72; 7%) and SSAs for all other causes (n = 12,378; 13%). Similarly, readmission within 30 days for the same diagnosis were more common among children with afebrile convulsions (n = 141; 10.5%) compared to both febrile convulsions (n = 42; 4%) and SSAs for all other causes (n = 5976; 6.5%).

## Qualitative interviews

Convulsions were discussed by 12 of the 41 professionals from our wider qualitative dataset, and an additional seven ESNs recruited specifically to share their experiences of supporting families in convulsion scenarios and suggest possible explanations for SSAs. Table 4 provides sample characteristics for 19 professionals contributing data for the analysis of convulsions. Of the 21 parents with experience of SSAs in the FLAMINGO study (sample characteristics detailed elsewhere [15]), four mothers had experience of SSA for convulsions pre-Covid for children aged under a year to 10 years and living in SIMD 1, 2, 4 and 5 across three Health Boards.

Findings from professional interviews are presented under three overarching themes (Fig 1), complemented by experiences of four parents.

**Theme 1 - Anxiety and panic.** Professionals and parents perceived parental anxiety and panic to be the driver for accessing care, especially from the emergency services. Parents referred to panicking when observing their child having a convulsion, whether it was the first or subsequent time. One parent explained how their panic was heightened when their child's febrile convulsion occurred outside usual GP opening hours. This was their reason for contacting the emergency service 999 phone line that triggers an ambulance attendance, whereas

**Table 4. Characteristics of interview participants (n = 19).**

| Professionals (n = 19) | |
|---|---|
| GP | 2 |
| Advanced Nurse Practitioner (ANP) | 1 |
| Consultant Paediatrician | 3 |
| Staff Nurse | 2 |
| Consultant Acute/Emergency Medicine | 2 |
| Senior Charge Nurse | 1 |
| Paediatric Advanced Nurse Practitioner (PANP) | 1 |
| Epilepsy Specialist Nurse (ESN) | 7 |

during the day they would contact the GP (Table 5, quote 1). Where parents had previous experience of their child having a febrile convulsion, a rise in temperature or inability to control a rising temperature was enough to instil panic and lead to hospital attendance. In addition to perceiving parents' fear around convulsions to be a primary reason for the attending ED (Table 5, quote 2), professionals explained how parental anxiety also informed their decisions about admitting, even in situations where the convulsion had subsided and clinically an admission was deemed unnecessary (Table 5, quote 3). Finally, as an ESN explained, whilst

**Theme 1:**
Anxiety and panic

- Parents feel overwhelming feelings of anxiety and panic when their child has a convulsion, often contributing to referral, a hospital attendance, or a SSA.

**Theme 2:**
Reassurance, observation, and proactive planning

- Reassurance and ongoing support for parents through both the observation of their child (by relevant professional) and proactive discharge planning to manage future convulsions and, if indicated, further investigations.

**Theme 3:**
Care from the right person, in the right place, at the right time

- Care pathways for acute convulsions that ensure care is provided by a professional with knowledge and expertise of convulsions in children, in the most appropriate setting and with timely access are required.

**Fig 1. Overarching themes related to experiences of unscheduled hospital attendance or SSA of children with convulsions.**

**Table 5. Exemplar quotations from interview participants.**

| Theme | Quote Number | Quotation | Participant |
|---|---|---|---|
| **Anxiety and panic** | 1 | "Because it's always been out of hours [when convulsion occurred] straight away I've just panicked and phoned 999. But I definitely think if it happened during the day, I would have let the situation calm down because I know what to do, they've told me what to do, and then just phone the GP for advice really." | Parent P011 |
| | 2 | "I think for some there is an absolute fear, when families watch their children have seizures they think their children are dying and if it's the first time they've ever seen that then that is more likely to be the reason they attend ED more than anything else." | Epilepsy Specialist Nurse P036 |
| | 3 | "For example, if you have a child who has a febrile convulsion, simple febrile convulsion, who makes a full recovery, you know, there's no reason why they can't go home but if the parents are completely shocked by it, which most parents are, yeah parental anxieties play a huge part [in admitting the child to hospital]." | Consultant P014 |
| | 4 | "For children with epilepsy. . .there are parents who feel safe once you've had a conversation with them, there are others who are hugely anxious and it wouldn't matter what you said to them, they would come to hospital regardless. To be honest, I think once they've met us a few times they're okay about that, they don't do it anymore. And then there are families who we train to use emergency buccal midazolam at the point of need. . .And some of the parents, they said it wouldn't matter, they're too anxious to use it, they're too scared and they would never not come to hospital." | Epilepsy Specialist Nurse P036 |
| **Reassurance, observation, and proactive planning** | 5 | "If it's going to get your child looked over by a professional, that's going to advise you on what's happened and the chances of this happening again and what to do. I think if I hadn't went there [to the ED] I would have probably started worrying what happens if it is something else and I don't know? But I think to get that reassurance, I wouldn't say it's an inconvenience yes because you're up during the night, you're exhausted then you miss school the next day if you've got other kids, but I don't think it's wasted at all." | Parent P011 |
| | 6 | "In something like, you know, a febrile convulsion, if it's parents who have had other kids that have had them and they are more relaxed about it, they might get home from the ED, but if it's first-time parents and they're very disconcerted by what they have seen, they might need to stay in for a bit of reassurance from the nurses." | Consultant Paediatrician P040 |
| | 7 | "If it's their first seizure, parents can obviously be really freaked out by it, but if it's someone that's known to have seizures and they've ended up in the ED, often the parent will tell you when they're ready to go. Yeah, most of the ones that I would keep overnight would be the ones that had a really long seizure or there are atypical features to it or there has been a history of a head injury or something like that, but I would say most, the vast majority of them, have their seizure in the community, they come in [to the ED], they feel back to their normal selves and then you send them away home with, if it's a febrile convulsion, you send them away with worsening advice, if it's a convulsion without fever or without infection, then you usually would follow them up in clinic after a few weeks." | Doctor P043 |
| | 8 | "Yeah, there was a leaflet given and it just tells you who to contact. If you've got concerns you can phone direct straight to the ward rather than going through the NHS24 thing or having to go through a switchboard, it was a direct line, any worries pop back up and then obviously they had a leaflet on febrile convulsions, so it told you everything, what the signs are etc." | Parent P011 |
| | 9 | "To be honest we were okay because we'd been reassured, while we'd been there all the morning and for most of the night before, we were kind of reassured that because somebody was finally paying attention, we knew it [convulsions] was going to be dealt with." | Parent P004 |
| | 10 | "I think [parents] still have this fear of febrile convulsion, you know, I think they still think that if their child's temperature is too high then they're going to have a fit. I think to them fever signifies that it's a serious illness, you know, so I think that's the reason why fever is one of the most concerning symptoms for parents." | Community Nurse P028 |
| | 11 | "We developed a febrile convulsion leaflet which families get, and it gives them information about temperature management and management of seizures. We also include febrile convulsions in our epilepsy awareness education sessions [to nurseries and schools] and they have started supplying information leaflets as part of their resources to families. They've started posting the leaflets on their wellbeing boards in the nursery, so again, it's been a way to filter some of the information out and down." | Epilepsy Specialist Nurse P048 |

*(Continued)*

**Table 5.** (*Continued*)

| Theme | Quote Number | Quotation | Participant |
|---|---|---|---|
| **Care from the right person, in the right place, at the right time** | 12 | "I think [child's] temperature was borderline 39°C so I phoned NHS24 who were very busy at the time which I totally get, who asked me to continue doing what I was doing, and they would get a nurse or doctor to call me. I had to call them back within that hour…[child's] temperature then went over 39 and I didn't want to get to 40. So, we made a decision as a family to take [child] straight to the ED." | Parent P001 |
| | 13 | "I know some families have got really frustrated with the NHS24 system because it takes so long [to speak to a HP]." | Epilepsy Specialist Nurse P041 |
| | 14 | "They [ED] were very, very busy at the time so I wasn't seen until half two in the morning …Granted they were very, very busy but they still checked up on [child] regardless of us being in the waiting room." | Parent 001 |
| | 15 | "I would definitely listen to my instincts with it and I would still, well depending on what happened, I would still phone 111 [NHS 24] for initial advice, but if I don't agree with what they said, if I thought anything was seriously wrong, I would just take [child] straight up the hospital." | Parent P004 |
| | 16 | "Within the space of five minutes all of this had disappeared and [child] was completely back to normal. So I go the phone call back from NHS24 and I said to them 'don't know what happened…she's now completely back to normal, all her responses are all exactly as they should be, I don't know what happened but I know something has?' and I basically got told it was probably a viral response and not to worry about it. And I was like 'yes, but [child]'s not displaying any viral activity, there's no raised temperature, there's no raised heart rate, there's absolutely nothing that would go with a viral response that would cause that' and I basically got told ' it's one of those things and I wouldn't worry about it too much but if you are concerned phone your GP in the morning'." | Parent P004 |
| | 17 | "It would have been quite useful to have been able to go on a video call with them [NHS24] and show them [symptoms]. I think it would be good for NHS24 as well because I think it would help them assess the situation a lot more clearly, and for the clinical teams as well to have access to video conferencing." | Parent P005 |
| | 18 | "So, families can send us in videos of their child having convulsions and we can look at these videos and we can liaise with the family through this system and we can tell them 'yes, we that that's this think or that's nothing' and that's a huge benefit because it stops children having to go into hospital and it also stops families having to trail up to hospital clinics to show us videos on phones, so that's been a huge breakthrough for us." | Epilepsy Specialist Nurse P035 |
| | 19 | "The paramedic had panicked me more and said 'we think this could be epilepsy' but there was no basis to say that, so immediately I was like 'oh okay what's going on, this is completely different?'. So [paramedic] had made a judgement without really knowing anything like, there was no tests that could be done at that point, [paramedic] just says 'oh we think this could be epilepsy, we really need to get to hospital quickly', so it was more an urgency and it was more panicked, whereas the first time [child had a febrile convulsion] it was very calm, the paramedics were helping me to remain calm, it was just completely different." | Parent P011 |
| | 20 | "I think the seizure management plans are a key part of preventing children from having to come to hospital. So I think if we had more nursing resource we'd be able to spend more time on the seizure management plans. For example, we might put on that plan that an ambulance doesn't need to be called, whereas schools and nurseries and respite centres often have a far lower threshold for calling an ambulance than families would and you can see why, they've got responsibility for a child in their care. So we often put, if we're sending our seizure management plans, that an ambulance doesn't need to be called and very clearly what steps can be taken before an ambulance should be called." | Epilepsy Specialist Nurse P045 |
| | 21 | "They all get a 48-hour open access, so what that means is that we give them our phone number with 48 hour open access details on it and instructing the parents that if they feel that their child is becoming worse and they're concerned then give us a phone back and we'll either given them advice over the phone or get them to come back in and see us." | Doctor P007 |

(*Continued*)

**Table 5.** (Continued)

| Theme | Quote Number | Quotation | Participant |
|---|---|---|---|
| | 22 | *"Open access came in quite a number of years ago, I'd say about five or six years again that it came in, and we found actually that it dropped our re-admission rate, surprisingly, we actually thought it would increase it but it actually dropped it and I think it's just because it's reassuring for the parents to know that they've got somewhere to come to if need be, some things we can manage over the phone so it ends up stopping them from re-attending. Yeah, so it has actually helped and it's quite reassuring for them."* | Doctor P008 |
| | 23 | *"So when we get a phone call from a family, what we do is ask a number of set questions and we then have to make the judgement and assessment of what can be managed over the phone or via a home visit. There are some situations that you can't safely manage over the phone so we might advise them to bring the child into hospital. Now probably our role is to try and manage as much at home to try and stop them need to come into hospital but quite often we do have to say 'listen, you need to bring them in, we need to get them reviewed'. So, we might actually be the one to tell them to come into the hospital to get admitted, whether or not that's for a period of observation or to change treatment."* | Epilepsy Specialist Nurse P035 |
| | 24 | *"We've got I think a reasonable policy of offering these families [children with complex and long term care needs] direct admission or direct access at least to advice, so they can phone and if they don't speak to the consultant that they know best or one of the consultants who are looking after them, they have access to the registrar. I think for those individuals it's very difficult for a more inexperienced member of staff, a junior doctor or nurse, to offer appropriate advice, but for the registrars it's an important part of their skills and ability to assess and offer advice to these families who we give, as I say, often open access to make contact. And I think it just limits the hassle for these families, and it's even worse for these youngsters, to go through out of hours because the out of hours team don't know them and they are complex and all the rest of it. So, in our unit anyways, open access for children with complex health issues is a reasonable common occurrence."* | Consultant Paediatrician P005 |

some parents can manage their child's convulsions with confidence, others find the situation too anxiety-provoking and are scared, for example, to administer emergency seizure medications, regardless of whether they have been previously supported and trained by ESNs to do so. For these parents, the preferred option would always be to attend hospital (Table 5, quote 4).

**Theme 2 - Reassurance, observation, and proactive planning.** Parents described their need for reassurance and ongoing support from professionals to manage the anxiety and panic experienced following a convulsion. The distressing nature of watching their child have a convulsion, often coupled with uncertainties around its cause and whether it might happen again, led parents to seek reassurance from professionals. When asked to share their views on the appropriateness of the decision to admit their child to hospital following a convulsion, parents referred to it being effective in reducing their worry, and that they considered this to be more important than any potential inconveniences related to a SSA (Table 5, quote 5). This concurs with the experience of professionals who acknowledged that whilst the clinical features of convulsions are the primary factor informing admission decisions, with those experiencing atypical or complex convulsions more likely to be admitted, a period of observation in hospital or a SSA is often necessary in cases of simple febrile convulsions to reassure parents (Table 5, quotes 6 and 7). Following any presentation with convulsions, professionals explained how the provision of safety netting advice and information leaflets to parents was an important intervention. This was confirmed by parents who described how prior to being sent home, they received instructions from hospital staff around what to do if another convulsion happens and what actions they should take if they have any concerns and found this to be beneficial and supportive (Table 5, quote 8).

Insights into the quantitative data linkage findings on the relatively high proportion of re-admissions for the same diagnosis when compared to all SSAs were shared by one parent whose child had repeated ED visits for frequent atypical convulsions, who was awaiting a

formal diagnosis. This parent felt reassured and supported only when hospital staff arranged hospital admission for additional diagnostic tests and assessments (Table 5, quote 9). ESNs highlighted the need for better continuity of care for children experiencing recurrent convulsions while they await referral to specialist services. ESNs suggested that in addition to discharge with safety netting advice, earlier follow up support from ESNs in the community could help prevent repeated ED attendances in situations where parents are seeking reassurance.

Several professionals emphasised that fevers with convulsions are concerning symptoms for parents. They reinforced the need to continue raising awareness and promoting safe management of fevers in children amongst parents, nurseries, schools and the wider public (Table 5, quotes 10 and 11). Professionals recommended that proactive planning in terms of further health education, reassurance and relieving parental fears may help empower parents to better cope if their child has a further convulsion with or without a fever and safely reduce some SSAs.

**Theme 3 - Care from the right person, in the right place, at the right time.** One parent whose child had a convulsion occurring outside GP opening hours described contacting the NHS24 service as their initial triage point, as is recommended UK urgent care guidance. However, the lengthy wait to speak to a professional via the NHS24 telephone service led to their decision to attend ED directly. In situations where parents are asked to wait for a doctor or nurse to call them back, any changes in their child's symptoms or increasing concern during that waiting time was often the catalyst for attending ED directly (Table 5, quote 12). Similarly, ESNs described parental expectations of timely access to healthcare during an acute convulsion in the community. They explained how parents often share their frustration with using the NHS24 pathway to access advice due to the long waits to speak with a professional (Table 5, quote 13). Parents could wait at varied points between home to the ED, including waiting for an NHS24 call back, for an ambulance, to be triaged, or to see a doctor with urgent assessment of paramount importance to families (Table 5, quote 14). Despite descriptions of the convulsions subsiding quickly and before the child and family had arrived at the ED, being seen by a professional with expertise of managing convulsions in children overrides any perceived inconvenience of attending hospital.

Parents described situations where there was a difference or disagreement in their own assessment of their child following a convulsion and the assessment of healthcare staff. If parents had concern that the NHS24 call handler or nurse was not accepting their assessment of the situation, they would still attend the ED (Table 5, quote 15). This has relevance in situations where there is a potential to discount rare but important symptoms or when triage is being undertaken by a someone without specialist knowledge of convulsions and is complicated in situations where the convulsion has resolved, and the child is no longer displaying symptoms (Table 5, quote 16). ESNs and a parent (Table 5, quote 17) described how video technology and telehealth can assist in the triage and care of children experiencing convulsions given their often transient and intermittent nature. Professionals positively described how video technology has rapidly accelerated during the COVID-19 pandemic where the use of videos recorded by parents or carers proved beneficial in communicating remotely with professionals when lockdown and social distance measures were in place. Several ESNs proposed further embedding this approach in practice (Table 5, quote 18).

The impact of receiving conflicting or inconsistent messages from professionals, particularly those encountered prior to attending hospital, was reported as an additional source of anxiety and unnecessary stressor, which, at times, further reinforced parents' decisions to seek emergency hospital care (Table 5, quote 19). Furthermore, lack of guidance and advice around management of convulsions in the community was considered by professionals to be another

potential contributor to SSAs. This could include, for example, school staff who opt to call an ambulance if a child has a convulsion at school despite a documented convulsion management plan with clear direction on immediate interventions to take following an acute convulsion in place (Table 5, quote 20).

Participants explained how having 'open access' as part of safety netting following a SSA for convulsions encouraged parents with concerns or changes in their child's condition over the first 48 hours, to contact the hospital ward or unit directly and arrange to be seen again if required (Table 5, quotes 8 and 21). This approach provided parents with direct access to staff who were familiar with the child's case and facilitated continuity of care. Open access may also prevent some readmissions, as suggested by one professional (Table 5, quote 22). ESNs explained how their role is to educate and empower families to manage their child's convulsions at home. However, in situations where the child requires further assessment or observation, having this specialist pathway where children are under the care of ESNs can allow them in some cases to bypass other referral routes and have the child seen in hospital directly (Table 5, quote 23). This is similar to offering direct access for children with complex conditions and care needs, which, as described by a consultant, is a reasonably common occurrence (Table 5, quote 24).

## Discussion

The main quantitative finding was that many characteristics of SSA for convulsions were different to those for all SSAs (see Table 3). Notably a greater proportion of SSAs are referred from ED when compared to all causes of SSAs. More than one referral source was contacted prior to 37% of SSA for convulsions. The frequency of readmissions for afebrile convulsions is twice as many as for SSAs for other causes with 10% readmitted within a month. The main qualitative findings are that timely access to, and reassurance from, health professionals to relieve parental anxiety and panic that the situation is life threatening are drivers of professional and parental behaviours. This can be the impetus for parents to attend hospital directly even if the convulsion has stopped, particularly when GP practices are closed. Collectively, the mixed method data suggest convulsions remain a frequent reason for SSAs in children and identify a requirement for further guidance for managing convulsions across pre and post hospital pathways.

Our qualitative findings around the fear and anxiety some parents experience following their child's convulsion provide some explanation for the ED being a primary referral source for SSAs and is consistent with existing evidence indicating parental anxiety and inadequate education around convulsion care are factors leading to urgent access to health services for children experiencing afebrile convulsions [19, 20]. Febrile convulsions are also well documented as a source of fear and anxiety for parents, especially concerning the potential dangers, and are a reason for attending the ED to seek reassurance [21–23]. Not all parents experience anxiety and many manage their child's convulsions with confidence, yet our findings indicate a requirement to address this issue of panic and resultant urgent ED attendance.

Westin and Sund Levander [21] found knowledge of febrile convulsions to be lacking among parents prior to their child's first febrile convulsion episode, and propose it be included in anticipatory advice given to all new parents. Similarly, whilst managing fever is common knowledge for health professionals, it may not be as readily known or accessible to parents [24]. Despite being published nearly 25 years ago, Kai's [24] findings remain pertinent today. Importantly, it resonates with recommendations for a safety netting approach to raise parents' awareness of febrile convulsions and to proactively provide health education around fever in infants and young children in our study.

Our qualitative findings are consistent with published research regarding the value of ESNs [25–27], the associated increase in hospital admissions when families lack access to such specialist support [28] and a recent UK-based national audit recommending continued expansion of the paediatric ESN service [29]. Moreover, our quantitative data on readmissions being more common in children with afebrile conditions further highlights the potential of specialist care to improve outcomes. Recommendations resulting from our analysis would include the provision of anticipatory guidance for parents on convulsions and additional community-based support to families following a SSA which may help safely avoid future admissions. ESNs are ideally placed to contribute to this. The potential value of using home-based video technology to support assessment and diagnosis of convulsions should also be further explored. Such an approach would help ensure care from the right person, in the right place, at the right time, a key theme identified in our interview data.

Our quantitative data highlight that readmissions within 30 days with further afebrile convulsions are common whilst our interview data suggest that a proportion of children with known convulsions as part of wider chronic and complex long-term conditions have some level of direct or alternative access to paediatric care thus bypassing the ED. This continuity of care model has, anecdotally, had a positive impact on rates of readmission. Similar accounts of open access to paediatric wards within 48 hours following ED attendance or a SSA for febrile convulsions were highlighted in our interviews with parents as part of the wider safety netting provided to them. It is plausible that such a service could be reducing SSAs for this population and improving the quality of care experience for families, and thus warrants further consideration. Future research is warranted to verify the impact of open access to hospital wards for children with known convulsions on SSAs.

It is important to acknowledge the challenges and complexities convulsions in children can pose for health professionals, especially those in generalist roles and without specialist paediatric expertise [3]. Adopting a cautious approach by staff involved in community care or triage may be necessary to manage uncertainty and risk, and there are situations where hospital admissions are necessary and the appropriate outcome. However, our findings suggest there is potential for safely reducing some SSAs related to convulsions. This could be addressed by modifying existing [9] or developing new pre- and post-hospital evidence-based guidelines to manage childhood convulsions in the community and address parents' concerns. Future research would be of merit to investigate the impact of implementation of such guidelines and interventions on SSAs for convulsions.

## Strengths and limitations

This post-hoc exploratory mixed methods analysis was conceived after identifying notable differences for admissions due to convulsions, compared to other conditions resulting in SSAs, within both quantitative and qualitative data collected during the FLAMINGO study. The mixed method approach facilitated detailed observations through data linkage, complemented by insights from qualitative interviews, to better understand the complexities of care pathways leading to SSAs for children experiencing convulsions.

Some limitations related to our linkage of data are described in Dick et al. [13], the main limitation being incomplete GP data availability which prevented full examination of access to care prior to attending hospital; for example, whether care was accessed from GPs earlier in the day or on the previous day prior to attending the ED following their child's convulsion. We were, however, able to describe the referral source for admissions with convulsions among the subgroup where GP data were available. It is important to acknowledge that we do not know how representative this sample is from the limited GP data we had access to.

Nonetheless, future extensive GP data linkage studies could have merit in identifying whether there is potential to streamline pre-hospital access to specialist assessment, particularly for recurrent seizure, which could be resource efficient as well as a means of allaying parental anxiety in a timely manner.

Our qualitative sample would have benefitted from further convulsion histories, for example, children with complex neurological and neurodevelopmental conditions and children with both a new and established epilepsy diagnosis, in addition to the four children from the original FLAMINGO dataset. We had intended to recruit more parents of children experiencing a SSA for convulsions, however this was not possible within the project timeline and resources due to the COVID-19 pandemic and its resultant impact on our recruitment strategy and access to potential participants. Including ESN perspectives on the patient experiences they have heard partially mitigates this as ESNs play a key role in supporting the coordination and continuity of care between community and hospital settings for children following a first convulsion and ongoing epilepsy support. Their inclusion triangulated earlier parent and professional interview findings and provided unique insights. Greater family diversity in health literacy, cultural and social backgrounds would have further enhanced our qualitative sample.

Most of our qualitative interviews were conducted during the COVID-19 pandemic, when accessing urgent care and the flow within hospitals was disrupted, yet the quantitative data used in our analysis originated from pre-pandemic times. We asked interview participants to focus on pre-pandemic experiences of SSAs before eliciting reflections on any current or future impacts of the pandemic and this can be regarded as a strength by offering additional consideration of how the pandemic may impact urgent care pathways in future.

## Conclusions

This in-depth analysis of quantitative and qualitative data from FLAMINGO highlights the differences between SSA for convulsions and for other conditions complemented by clinicians' experience of caring for children with convulsions. Pre- and post-hospital convulsion pathways have the potential to safely streamline admissions and reduce readmissions which will improve parent experiences and reduce pressure on acute NHS services. Enhanced specialist nurses would play a key role in these care pathways and improve continuity of care.

## Supporting information

**S1 File. Interview topic guide.**
(DOCX)

**S2 File. List of pre-existing neurological conditions.**
(DOCX)

## Acknowledgments

The authors would like to extend our thanks to the parents and health professionals who participated in this study.

## Author Contributions

**Conceptualization:** Cari Malcolm, Pat Hoddinott, Richard Kyle, Philip Wilson, Emma France, Lorna Aucott, Stephen W. Turner.

**Data curation:** Emma King, Smita Dick, Stephen W. Turner.

**Formal analysis:** Cari Malcolm, Pat Hoddinott, Emma King, Smita Dick, Lorna Aucott, Stephen W. Turner.

**Funding acquisition:** Cari Malcolm, Pat Hoddinott, Richard Kyle, Philip Wilson, Emma France, Lorna Aucott, Stephen W. Turner.

**Investigation:** Cari Malcolm, Pat Hoddinott, Emma King, Philip Wilson, Stephen W. Turner.

**Methodology:** Cari Malcolm, Pat Hoddinott, Richard Kyle, Philip Wilson, Emma France, Lorna Aucott, Stephen W. Turner.

**Project administration:** Cari Malcolm, Emma King, Smita Dick, Stephen W. Turner.

**Supervision:** Stephen W. Turner.

**Validation:** Pat Hoddinott.

**Writing – original draft:** Cari Malcolm.

**Writing – review & editing:** Pat Hoddinott, Emma King, Smita Dick, Richard Kyle, Philip Wilson, Emma France, Lorna Aucott, Stephen W. Turner.

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
