## [Decision Letter · Decision Letter 0]

13 Dec 2023

PONE-D-23-34920Short-stay urgent hospital admissions of children with convulsions: a mixed methods exploratory study to inform out of hospital care pathways.PLOS ONE

Dear Dr. Malcolm,

Thank you for submitting your manuscript to PLOS ONE. After careful consideration, we feel that it has merit but does not fully meet PLOS ONE’s publication criteria as it currently stands. Therefore, we invite you to submit a revised version of the manuscript that addresses the points raised during the review process.

**ACADEMIC EDITOR:**
**-**please follow the comments of our expert reviewers ==============================

We look forward to receiving your revised manuscript.

Kind regards,

Prof. Dragan Hrncic, MD, PhD

Academic Editor

PLOS ONE

Journal Requirements:

"This study was funded by the Chief Scientist Office Scotland (HIPS/18/09)."

5. Please note that in order to use the direct billing option the corresponding author must be affiliated with the chosen institute. Please either amend your manuscript to change the affiliation or corresponding author, or email us at plosone@plos.org with a request to remove this option.

Reviewers' comments:

Reviewer's Responses to Questions

**Comments to the Author**

1. Is the manuscript technically sound, and do the data support the conclusions?

Reviewer #1: Yes

2. Has the statistical analysis been performed appropriately and rigorously? 

Reviewer #1: Yes

3. Have the authors made all data underlying the findings in their manuscript fully available?

Reviewer #1: Yes

4. Is the manuscript presented in an intelligible fashion and written in standard English?

Reviewer #1: Yes

5. Review Comments to the Author

Reviewer #1: Thank you very much for this interesting article. I only have two comments to improve your study:

- Specify with more details the contribution of your work with respect the previous ones published by you. As you stated, "The main FLAMINGO study findings are published elsewhere [13-14]", and the contribution of this work should be emphasized better.

- Suggest which might be the future direction of your studies. For instance, what about the adoption of AI to support the stratification of your patients? This is a sector in which the literature (and the medicine) is going, and it might be interesting to highlight this opportunity (e.g., see Falavigna et al., 2019; Ippoliti et al., 2021).

Citation

Falavigna, G., Costantino, G., Furlan, R., Quinn, J. V., Ungar, A., & Ippoliti, R. (2019). Artificial neural networks and risk stratification in emergency departments. Internal and Emergency Medicine, 14, 291-299.

Ippoliti, R., Falavigna, G., Zanelli, C., Bellini, R., & Numico, G. (2021). Neural networks and hospital length of stay: an application to support healthcare management with national benchmarks and thresholds. Cost Effectiveness and Resource Allocation, 19(1), 1-20.

6. PLOS authors have the option to publish the peer review history of their article (what does this mean?). If published, this will include your full peer review and any attached files.

Reviewer #1: No

---

## [Author Response · Author response to Decision Letter 0]

11 Jan 2024

Manuscript PONE-D-23-34920 - Authors’ responses to reviewer comments:

Any changes or amendments the authors have made to the manuscript in line with reviewer comments can be identified by the use of yellow highlighting.

Reviewer 1 Comments and Author Responses

Thank you for your insightful and helpful comments on our manuscript. We have considered them in depth and would like to offer the following responses to each of the comments and suggestions proposed:

Comment 1 – Specify with more details the contribution of your work with respect to the previous ones published by you. As you stated, “The main FLAMINGO study findings are published elsewhere [13-14]”, and the contribution of this work should be emphasised better. 

Author Response 

On page 3 we have described the focus of the publications reporting the main FLAMINGO study findings (new text in yellow):

‘The main FLAMINGO study findings, published elsewhere, report on quantitative data linkage analysis of routinely acquired Scottish data to identify referral sources and characteristics for all urgent paediatric SSAs [13] and on qualitative interviews with 21 parents and 48 health professionals communicating their desired outcomes of pre-hospital urgent care for all causes of SSAs [14].

Comment 2 – Suggest what might be the future direction of your studies. For instance, what about the adoption of AI to support the stratification of your patients? This is a sector in which the literature (and the medicine) is going, and it might be interesting to highlight this opportunity (eg. See Falavigna et al., 2019; Ippoliti et al., 2021).

Citation:

Falavigna, G., Costantino, G., Furlan, R., Quinn, J. V., Ungar, A., & Ippoliti, R. (2019). Artificial neural networks and risk stratification in emergency departments. Internal and Emergency Medicine, 14, 291-299.

Ippoliti, R., Falavigna, G., Zanelli, C., Bellini, R., & Numico, G. (2021). Neural networks and hospital length of stay: an application to support healthcare management with national benchmarks and thresholds. Cost Effectiveness and Resource Allocation, 19(1), 1-20.

Author Response –

Thank you for suggesting that we could indicate areas for future research. We have added suggestions on pages 25 and 26 as follows:

‘Future research is warranted to verify the impact of open access to hospital wards for children with known convulsions on SSAs.’

‘Future research would be of merit to investigate the impact of implementation of guidelines and interventions on SSAs for convulsions.’

We enjoyed reading and discussing these two papers (Falavigna et al. 2019; Ippoliti et al. 2021) and agree there is potential in adopting AI to support the stratification of patients within the ED and manage patient flow. It has relevance to our wider FLAMINGO project, and we will refer to back these citations when writing future publications. However, with regard to this paper, the value of adopting a mixed method approach as we have done in this study, highlights the potential to intervene and provide care to children with convulsions before they arrive at the ED. Our findings support the consideration of pre-hospital pathways for paediatric convulsions and therefore the discussion around potential use of AI is not relevant here.

We have also taken the opportunity to correct minor typographical errors or omissions indicated in yellow highlighter throughout.

Yours faithfully, 

Cari Malcom on behalf of the FLAMINGO team

---

## [Decision Letter · Decision Letter 1]

29 Jan 2024

PONE-D-23-34920R1Short-stay urgent hospital admissions of children with convulsions: a mixed methods exploratory study to inform out of hospital care pathways.PLOS ONE

Dear Dr. Malcolm,

Thank you for submitting your manuscript to PLOS ONE. After careful consideration, we feel that it has merit but does not fully meet PLOS ONE’s publication criteria as it currently stands. Therefore, we invite you to submit a revised version of the manuscript that addresses the points raised during the review process.

 Please submit your revised manuscript by Mar 14 2024 11:59PM. If you will need more time than this to complete your revisions, please reply to this message or contact the journal office at plosone@plos.org. Please include the following items when submitting your revised manuscript:A rebuttal letter that responds to each point raised by the academic editor and reviewer(s). You should upload this letter as a separate file labeled 'Response to Reviewers'.A marked-up copy of your manuscript that highlights changes made to the original version. You should upload this as a separate file labeled 'Revised Manuscript with Track Changes'.An unmarked version of your revised paper without tracked changes. You should upload this as a separate file labeled 'Manuscript'.

We look forward to receiving your revised manuscript.

Kind regards,

Dragan Hrncic

Academic Editor

PLOS ONE

Additional Editor Comments:

**ACADEMIC EDITOR:** - do respond in due time 

Reviewers' comments:

Reviewer's Responses to Questions

**Comments to the Author**

1. If the authors have adequately addressed your comments raised in a previous round of review and you feel that this manuscript is now acceptable for publication, you may indicate that here to bypass the “Comments to the Author” section, enter your conflict of interest statement in the “Confidential to Editor” section, and submit your "Accept" recommendation.

Reviewer #1: (No Response)

2. Is the manuscript technically sound, and do the data support the conclusions?

Reviewer #1: Partly

3. Has the statistical analysis been performed appropriately and rigorously? 

Reviewer #1: Yes

4. Have the authors made all data underlying the findings in their manuscript fully available?

Reviewer #1: No

5. Is the manuscript presented in an intelligible fashion and written in standard English?

Reviewer #1: Yes

6. Review Comments to the Author

Reviewer #1: I believe this article did not consider the main comment suggested in the first round of this review, which is fundamental to satisfy the key criteria of this journal, i.e., whether the study presents the results of original research (i), and whether results reported have not been published elsewhere (ii).

Indeed, authors do not explain properly how this study differs from the previous ones (published by authors on the same study), and they do not specify the contribution of this work with respect the current (published) knowledge.

7. PLOS authors have the option to publish the peer review history of their article (what does this mean?). If published, this will include your full peer review and any attached files.

Reviewer #1: No

---

## [Author Response · Author response to Decision Letter 1]

12 Feb 2024

Manuscript PONE-D-23-34920 - Authors’ responses to reviewer comments:

Reviewer 1 Comments and Author Responses

Thank you for your insightful and helpful comments on our manuscript. We have considered them in depth and would like to offer the following response:

Comment 1 – I believe this article did not consider the main comment suggested in the first round of this review, which is fundamental to satisfy the key criteria of this journal, ie., whether the study presented the results of original research (i), and whether results reported have not been published elsewhere (ii). Indeed, authors do not explain properly how this study differs from the previous ones (published by authors on the same study), and they do not specify the contribution of this work with respect to the current (published) knowledge.

Author Response 

The authors regret that the reviewer’s substantive comments were not appropriately addressed and are grateful for the opportunity to respond to them in this revised version. We are confident this manuscript meets the essential PLOS ONE criteria, that being the communication of original research that has not been previously published in the peer-reviewed literature. We hope that the justification and amendments below clearly articulate this.

The Flow of Admissions in Children and Young People (FLAMINGO) project was designed to explore urgent short stay hospital admissions for acutely unwell children, namely their characteristics and referral patterns. Analysis of this data drew our attention to convulsions as a focus of interest given their higher prevalence in Emergency Department (ED) attendance and referral in comparison to the nine other clinical presentations included in the data. This informed our decision to undertake the following novel analyses: (i) stratification of quantitative data by febrile and afebrile convulsions, also presenting time of admission and details of readmissions (ii) further qualitative interviews with parents, professionals, including Epilepsy Specialist Nurses was undertaken to explore their experiences of Short Stay Admissions (SSAs) for children with afebrile and febrile convulsions (iii) thematic analysis of interviews. Our present paper arose from observations made in an earlier paper and, as explained above, presents novel results. 

The novel contributions of this work with respect to the existing and published evidence base is communicated in detail within the discussion section, particularly highlighting the value of using a mixed methods approach which facilitates a better understanding of the complexities of care pathways leading to SSAs for children experiencing convulsions. 

The results reported in this paper have not been published elsewhere. Below is a list of each FLAMINGO paper and a summary of any convulsion data published in them:

Dick S, Kyle R, Wilson P et al. Insights from and limitations of data linkage studies: analysis of short-stay urgent admission referral source from routinely collected Scottish data. Archives of Disease in Childhood. 2023 Apr 1; 108(4): 300-6. Convulsions are mentioned nine times, in context of the referral source in comparison to children presenting with other conditions. There is no stratification or presentation of data by febrile of afebrile status, readmission rates or time of admission.

King E, Dick S, Hoddinott P et al. Regional variations in short stay urgent paediatric hospital admissions: a sequential mixed-methods approach exploring differences through data linkage and qualitative interviews. BMJ Open. 2023 Sep 1; 13(9): e072734. Convulsions are referred to three times only in the context of differences in SSA rates between regions. 

Malcolm C, King E, France E, et al. Short stay hospital admissions for an acutely unwell child: a qualitative study of outcomes that matter to parents and professionals. PLOS ONE. 2022 Dec 16; 17(12): e0278777. The is no reference to convulsions in this paper. 

King E, France E, Malcolm C, Kumar S, Dick S, Kyle RG, Wilson P, Aucott L, Turner S, Hoddinott P. Identifying and prioritising future interventions with stakeholders to improve paediatric urgent care pathways in Scotland, UK: a mixed methods study. BMJ Open. 2023 Oct 1; 13(10):e074141. Convulsions are mentioned in the context of identifying convulsion as a presentation with a potential pathway of care – there is no convulsion data or findings presented or discussed. 

Changes to manuscript

Any changes or amendments the authors have made to the manuscript in line with reviewer comments can be identified by the use of yellow highlighting:

The Flow of Admissions in Children and Young People (FLAMINGO) project was undertaken in response to the growing number of paediatric urgent hospital admissions in the UK [4-11], explained largely by rising SSAs [4,12]. This exploratory sequential mixed-methods study aimed to improve understanding of SSAs in Scotland through linkage of national databases to identify referral pathways and characteristics of SSAs. Qualitative interviews with professionals were complemented by parent experiences and Public Patient Involvement. The main FLAMINGO study findings, published elsewhere, report on quantitative data linkage analysis of routinely acquired Scottish data to identify referral sources and characteristics for urgent paediatric SSAs arising from a range of common conditions including, but not limited to, viral illness, asthma and bronchiolitis [13-14] and on qualitative interviews with 21 parents and 48 health professionals communicating their desired outcomes of pre-hospital urgent care for these urgent SSAs [15]. Data linkage findings from the main FLAMINGO dataset indicated a high prevalence of SSAs for convulsions presenting at the emergency department (ED) [13] which, given the lack of recently published evidence in this area, warranted further detailed investigation. Previous FLAMINGO papers [13-16] did not specifically report the characteristics of SSAs related to convulsions or parental and professional experiences of these admissions. Therefore, additional quantitative analysis and qualitative research involving the recruitment of Epilepsy Specialist Nurses (ESN)s was undertaken. This paper communicates novel findings from this research to identify learning for improvement in care pathways and development of potential interventions focused on safely reducing urgent SSAs for convulsions and improving care pathways, by analysing (i) characteristics of SSAs for children with afebrile or febrile convulsions, and (ii) qualitative interviews with professionals and parents with experience of childhood convulsions.

---

## [Decision Letter · Decision Letter 2]

11 Mar 2024

Short-stay urgent hospital admissions of children with convulsions: a mixed methods exploratory study to inform out of hospital care pathways.

PONE-D-23-34920R2

Dear Dr. Malcolm,

We’re pleased to inform you that your manuscript has been judged scientifically suitable for publication and will be formally accepted for publication once it meets all outstanding technical requirements.

Kind regards,

Prof. Dr. Dragan Hrncic, MD, PhD

Academic Editor

PLOS ONE

Additional Editor Comments (optional):

Reviewers' comments:

Reviewer's Responses to Questions

**Comments to the Author**

1. If the authors have adequately addressed your comments raised in a previous round of review and you feel that this manuscript is now acceptable for publication, you may indicate that here to bypass the “Comments to the Author” section, enter your conflict of interest statement in the “Confidential to Editor” section, and submit your "Accept" recommendation.

Reviewer #1: All comments have been addressed

2. Is the manuscript technically sound, and do the data support the conclusions?

Reviewer #1: Partly

3. Has the statistical analysis been performed appropriately and rigorously? 

Reviewer #1: Yes

4. Have the authors made all data underlying the findings in their manuscript fully available?

Reviewer #1: Yes

5. Is the manuscript presented in an intelligible fashion and written in standard English?

Reviewer #1: Yes

6. Review Comments to the Author

Reviewer #1: The new version has been improved according to expectations, and it can be published on the journal.

7. PLOS authors have the option to publish the peer review history of their article (what does this mean?). If published, this will include your full peer review and any attached files.

Reviewer #1: No

---

## [Editor Report · Acceptance letter]

21 Mar 2024

PONE-D-23-34920R2 

PLOS ONE

Dear Dr. Malcolm, 

I'm pleased to inform you that your manuscript has been deemed suitable for publication in PLOS ONE. Congratulations! Your manuscript is now being handed over to our production team.

Kind regards, 

on behalf of

Professor Dragan Hrncic 

Academic Editor

PLOS ONE